**Data Availability Statement:** All relevant data are within the paper and its Supporting Information files.

**Funding:** The author(s) received no specific funding for this work.

# A new clinical prognostic nomogram for liver cancer based on immune score

**Qinyan Shen**[ID]*, **Guinv Hu, JinZhong Wu, Liting Lv**

Department of Surgical Oncology, Affiliated Dongyang Hospital of Wenzhou Medical University, Dongyang, Zhejiang, China

* sqy198977@163.com

## Abstract

### Background

Increased attention is being paid to the relationship between the immune status of the tumor microenvironment and tumor prognosis. The application of immune scoring in evaluating the clinical prognosis of liver cancer patients has not yet been explored. This study sought to clarify the association between immune score and prognosis and construct a clinical nomogram to predict the survival of patients with liver cancer.

### Methods

A total of 346 patients were included in our analysis datasets downloaded from The Cancer Genome Atlas (TCGA) dataset. A Cox proportional-hazards regression model was used to estimate the adjusted hazard ratios (HRs). A nomogram was built based on the results of multivariate analysis and was subjected to bootstrap internal validation. The predictive accuracy and discriminative ability were measured by the concordance index (C-index) and the calibration curve. Through the functional analysis of differential expression of genes with different immune scores, the target genes were screened out.

### Results

In comparison with patients with low immune scores, those with intermediate and high immune scores had significantly improved survival time [HR and 95% confidence interval (CI): 0.54 (0.30–0.97) and 0.51 (0.27–0.97), respectively]. The C-index for survival time prediction was 0.66 (95% CI: 0.60–0.71). The calibration plot for the probability of survival at three or five years showed good agreement between prediction by the nomogram and actual observations. The top 10 hub genes were *CXCL8*(chemokine (C-X-C motif) ligand 8), *SYK*(spleen tyrosine kinase), *CXCL12*(chemokine (C-X-C motif) ligand 12), *CXCL10* (chemokine (C-X-C motif) ligand10), *CXCL1*(chemokine (C-X-C motif) ligand1), *CCL5*(chemokine (C-C motif) ligand 5), *CCL20*(chemokine (C-C motif) ligand 20), *LCK*, *CXCL11* (chemokine (C-X-C motif) ligand 11), *CCR5*(chemokine (C-C motif) receptor 5). More importantly, we found that the high expression of *CXCL8* and *CXCL1* were related to the prognosis.

**Competing interests:** The authors have declared that no competing interests exist.

## Conclusions

High and/or intermediate immune scores are significantly correlated with better survival time in patients with liver cancer. Moreover, nomograms for predicting prognosis may help to estimate the survival of patients. We also propose that *CXCL8* and *CXCL1* may be a potential therapeutic target for liver cancer treatment.

## Introduction

Liver cancer is one of the most frequently diagnosed malignancies and the fourth leading cause of death from cancer worldwide. On the basis of annual projections, the World Health Organization estimates that more than one million patients will die from liver cancer in 2030 [1]. According to 2019 cancer statistics, the incidence of liver cancer is rising faster than that of any other cancer in both men and women in the United States [2]. Chronic hepatitis C virus (HCV) or chronic hepatitis B virus (HBV) infections, alcohol abuse, and nonalcoholic steatohepatitis are the main risk factors for this disease [3]. Despite the rapid progress being made in surgical techniques, which are the primary therapies for liver cancer, in combination with adjuvant chemotherapy or radiotherapy, patients with liver cancer frequently relapse following liver resection [4]. However, 70% to 80% of patients cannot benefit from surgery because they are diagnosed at an advanced stage and are only eligible for palliative care. In recent years, preclinical researches and clinical trials have offered many opportunities for the development of liver cancer treatment. Immune therapeutic strategies have been proven safe and effective [5]. Unlike conventional cancer therapies, immunotherapeutic approaches do not directly target tumor cells; instead, they target the patient's immune system or the tumor microenvironment (TME) [6]. A variety of strategies have been explored: cytokine administration, cancer vaccines, adoptive cellular therapy and immune checkpoint blockade (ICB) [7]. Among which, ICB have been subject to cancer immunotherapy due to its promising outcomes across multiple advanced solid malignancies, including hepatocellular carcinoma (HCC). The key mechanism of action for ICB is to block the immune exhaustion or inhibitory pathways induced by chronic immune response against tumor antigen, in order to reactivate the antitumor immune response. PD-1, PD-L1, and CTLA-4 inhibitors are the most widely evaluated ICB therapies in clinical trials for HCC [8–13]. Multiple immunotherapeutic strategies have been tested in HCC, with some degree of success, particularly with immune checkpoint blockade (ICB). Despite the initial enthusiasm, treatment benefit is only appreciated in a modest proportion of patients. Challenges stay in identifying HCC patients who could best benefit from immunotherapy. Therefore, understanding the relationship between the immune system and prognosis is vital to effectively utilize promising immuno-oncology agents.

Tumor microenvironment (TME) cells are a vital element of tumor tissue. Recently, accumulating evidence has elucidated their clinicopathologic significance in predicting outcomes in various malignancies, including gastric cancer, ovarian cancer, neuroblastoma, and melanoma [14–17]. Notably, immune and stromal cells, two major components of non-tumor cell populations in the TME, have been identified as offering a prognostic assessment of the tumor [18,19]. Yoshihara et al. designed an algorithm based on gene expression signatures to estimate immune and stromal cells, as well as tumor purity, called the Estimation of Stromal and Immune cells in Malignant Tumor Tissues using Expression Data (ESTIMATE) [20]. ESTIMATE scores correlate with DNA copy number-based tumor purity across samples from 11 different tumour types, profiled on Agilent, Affymetrix platforms or based on RNA sequencing

and available through The Cancer Genome Atlas. The prediction accuracy is further corroborated using 3,809 transcriptional profiles available elsewhere in the public domain. The ESTIMATE method allows consideration of tumour-associated normal cells in genomic and transcriptomic studies. The ESTIMATE algorithm has since been adopted to assess many malignancies, such as lung cancer, breast cancer, prostate cancer, cholangiocarcinoma, glioblastoma, lung cancer, salivary duct carcinoma, and colon cancer [21–28], whereas the prognostic value of immune and/or stromal scores of liver cancer has not been sufficiently investigated.

Here, we comprehensively analyzed 346 liver cancer cases, with clinicopathologic characteristics and immune scores obtained from The Cancer Genome Atlas (TCGA), to evaluate the association of immune score with prognosis and to construct a clinical nomogram for predicting the survival of patients with liver cancer.

## Material and methods

### Materials

We used public data downloaded from the TCGA dataset for this research. TCGA (https://portal.gdc.cancer.gov) is currently the largest dataset available for the genomic analysis of tumors, including at least 200 kinds of cancer and associated clinical information as well as measurements such as DNA methylation and RNA sequencing. TCGA's clinicopathological information was downloaded from an open resource, which included the unique number of patients, age, tumor node metastasis (TNM) findings, tumor grade, status, and survival time. Immune scores were downloaded from the ESTIMATE website (https://bioinformatics.mdanderson.org/estimate), which provides researchers with scores for tumor purity, the level of stromal cells present, and the infiltration level of immune cells in tumor tissues based on expression data. This website is designed to view and download stromal, immune, and ESTIMATE scores for each sample across all TCGA tumor types and platforms.

### Data preprocessing

A total of 346 cases could be used for further analysis after the number of duplicates was excluded. The specific elimination analysis process is listed in (S1 Fig) as a flowchart. Each immune score corresponds to one patient.

### Statistical analysis

The cutoff point for immune score was obtained using X-tile 3.6.1 software (Yale University School of Medicine, New Haven, CT, USA), as described previously [29]. X-tile plots were conducted for the assessment of immune scores; this was expressed as the optimization of cutoff points based on outcome. Categorical data were analyzed using the chi-squared test or Fisher's exact test, and continuous variables were analyzed using the analysis of variance test or the Kruskal–Wallis H test for variables with an abnormal distribution and homogeneity of variance. Survival curves were constructed using the Kaplan–Meier method and were compared using the log-rank test. A multivariate Cox proportional-hazards regression model was used to identify the independent predictors of survival time. After the effects of age, tumor grade, TNM stage, and immune score were simultaneously considered, adjusted hazard ratios (HRs) and 95% confidence intervals (CIs) were estimated.

A total of 346 patients were divided into high and low immune score groups according to the immune score results. The differentially expressed genes (DEGs) were identified using the package limma in R version 3.6.2 (R Foundation for Statistical Computing, Vienna, Austria),

and the cutoffs were fold change $> 1$ and adjust $P < 0.05$. To assess the potential biologic functions of differentially expressed immune-related genes, Gene Ontology (GO) and Kyoto Encyclopedia of Genes and Genomes (KEGG) pathway enrichment analysis were performed by the cluster Profiler package in R. Functional categories with a adjusted P value $< 0.05$ or FDR $< 0.05$ were considered as significant pathways. The PPI network of DEGs was constructed according to information acquired using the STRING database (https://string-db.org/). To identify hub genes in the PPI network, we implemented maximal clique centrality analysis into cytoHubba (a Cytoscape plugin). Maximal clique centrality is a topological analytical method that effectively screens for hub genes. In addition, the expression of all the 10 genes was verified on the GEPIA (http://gepia.cancer-pku.cn/).

Nomograms were formulated based on the results of multivariate analysis using R software. These nomograms were subjected to 1000 bootstrap resamples for internal validation of the analyzed database. The performance of models for predicting prognosis was evaluated by calculating the concordance index (C-index). The value of the C-index was between 0.5 and 1.0, with 1.0 indicating the perfect ability to correctly discriminate the outcomes with the model and 0.5 indicating a random chance. Calibration of the nomogram for three and five years of survival was performed by comparing the observed survival with the predicted survival probability. All statistical tests were two-sided and p-values of less than 0.05 were considered to be statistically significant. Data compilations and descriptive statistics were performed using the IBM SPSS version 23 software program (IBM Corp., Armonk, NY, USA).

## Results

### Patients' characteristics

A total of 346 patients were included in our analysis datasets after data cleaning (for specific data preparation, see S1 Fig). The average age of patients was 58.79 years (standard deviation: 13.46 years, range: 16–85 years). Median immune scores of patients were −108.41 (range: −1209.20 to 2934.40, interquartile range: 419.55). The cutoff points for immune score were −786.40 and 268.70; thus, patients were subsequently subdivided into high, intermediate, and low immune score subgroups (X-tile plots are shown in S2 Fig). In total, the scores of 35 (10.12%) patients were lower than or equal to −786.4 (low immune score subgroup), 198 (57.23%) had scores between −786.4 and 268.7 (intermediate immune score subgroup), and 113 (32.66%) patients had scores greater than 268.7 (high immune score subgroup). The median survival time was 542.50 days (range: 0–3675 days). Table 1 presents the clinicopathologic features of the different subgroups according to immune score. As compared with the low immune score subgroup, the patients with intermediate and high immune scores tended to be stages II and III.

### Multivariate analyses for survival time

Results of the multivariate Cox proportional-hazards regression analyses are shown in Table 2 and S3 Fig, S4 Fig. In comparison with patients with low immune scores, those with intermediate and high immune scores had significantly improved survival time [HR and 95% CI: 0.54 (0.30–0.97) and 0.51 (0.27–0.97), respectively]. As expected, when compared with patients with stage I disease, those with stage III or IV disease had significantly poorer survival time (HRs and 95% CIs for stages II, III, and IV were 1.37 (0.82–2.31 2.71 (1.74–4.23), and 6.26 (2.08–18.78), respectively). As for the rest of the clinical characteristics, significant associations were not recognized.

**Table 1. Associations between clinicopathological features and immune scores in 346 liver cancer patients.**

| | | Immune scores | | | | |
|---|---|---|---|---|---|---|
| Characteristics | Total | Low | Medium | High | $\chi^2$ value | p-value |
| Sample sizes | 346 | 35 | 198 | 113 | | |
| Age (y) | | | | | 6.16 | 0.63 |
| < 40 | 32 | 5 | 17 | 10 | | |
| 40–49 | 38 | 2 | 25 | 11 | | |
| 50–59 | 93 | 13 | 47 | 33 | | |
| 60–69 | 112 | 8 | 67 | 37 | | |
| ≥ 70 | 71 | 7 | 42 | 22 | | |
| Tumor grade | | | | | 4.68 | 0.59 |
| G1 | 46 | 6 | 28 | 12 | | |
| G2 | 169 | 12 | 98 | 59 | | |
| G3 | 118 | 16 | 65 | 37 | | |
| G4 | 13 | 1 | 7 | 5 | | |
| Stage | | | | | 8.53 | 0.20 |
| Stage I | 171 | 12 | 96 | 63 | | |
| Stage II | 85 | 8 | 49 | 28 | | |
| Stage III | 84 | 14 | 49 | 21 | | |
| Stage IV | 6 | 1 | 4 | 1 | | |

p < 0.05; difference was statistically significant.

**Table 2. Multivariate analyses of survival time among liver cancer patients according to clinical characteristics and immune scores.**

| | survival time | |
|---|---|---|
| Characteristics | HR (95% CI) | p-value |
| Age | | |
| < 40 | 1 | |
| 40–49 | 1.12 (0.47–2.71) | 0.80 |
| 50–59 | 1.12 (0.52–2.41) | 0.78 |
| 60–69 | 1.15 (0.54–2.44) | 0.72 |
| ≥ 70 | 1.74 (0.82–3.69) | 0.15 |
| Tumor grade | | |
| G1 | 1 | |
| G2 | 1.17 (0.64–2.16) | 0.61 |
| G3 | 1.25 (0.66–2.34) | 0.50 |
| G4 | 2.44 (0.88–6.77) | 0.09 |
| Stage | | |
| Stage I | 1 | |
| Stage II | 1.37 (0.82–2.30) | 0.23 |
| Stage III | 2.71 (1.74–4.23) | < 0.001* |
| Stage IV | 6.26 (2.08–18.78) | 0.001* |
| Immune score | | |
| Low | 1 | |
| Medium | 0.54 (0.30–0.97) | 0.04* |
| High | 0.51 (0.27–0.97) | 0.041* |

* p < 0.05; difference was statistically significant.

## Differentially expressed genes and functional enrichment analysis

A total of 1122 genes (1041 upregulated and 81 downregulated) were identified as differentially expressed in high immune score groups compared with and low immune score groups. The 1122 differentially expressed immune-related genes were further analyzed by GO and KEGG analysis. GO analysis revealed that primary functional categories in the biological processes (BP) were T cell activation, leukocyte migration and leukocyte cell-cell adhesion. For cellular components (CC), the major enriched GO terms were external side of plasma membrane and collagen-containing extracellular matrix. The most enriched molecular function (MF) were receptor cytokine activity, chemokine activity and receptor ligand activity. (S5A Fig). KEGG pathway indicated that the differentially expressed immune-related genes were mainly involved in Cytokine-cytokine receptor interaction, Hematopoietic cell lineage and Chemokine signaling pathway. (S5B Fig). The results of Cytoscape showed that 33 genes were related to each other (S6A Fig). According to cytoHubba plugin's Degree ranking, the top 10 hub genes were *CXCL8*, *SYK*, *CXCL12*, *CXCL10*, *CXCL1*, *CCL5*, *CCL20*, *LCK*, *CXCL11*, *CCR5*(S6B Fig). In addition, we found that highly expressed *CXCL8* and *CXCL1* had a poor Overall survival (OS)(S7 Fig).

**Prognostic nomogram for survival time.**   The prognostic nomogram that integrated all considered independent factors for survival time is shown in S8 Fig. The C-index for survival time prediction was 0.66 (95% CI: 0.60–0.71). The calibration plot for the probability of survival at three and five years showed good agreement between prediction by the nomogram and actual observations (S9 Fig).

## Discussion

In the current study, we evaluated the prognostic significance of immune scores by using gene expression data from patients with liver cancer. After possible confounders were considered, we found that high and/or intermediate immune scores were significantly associated with the survival time of liver cancer patients. Importantly, we found that immune related genes *CXCL8* and *CXCL1* are related to prognosis. Meanwhile, we also constructed nomograms to easily predict the survival of patients with liver cancer.

The initiation and progression of liver cancer, including hepatocellular carcinoma and intrahepatic cholangiocarcinoma, are dependent on the tumor microenvironment. Immune cells are key players in the liver cancer microenvironment and conduct complicated crosstalk with cancer cells. The prognostic importance of immune cell infiltration has been recognized for different solid tumor types [21,22,30,31]. It has been previously reported that T- and B-cells are present in immune cell infiltrates of hepatocellular carcinoma (HCC) and that the degrees of tumor-infiltrating T- and B-cells correlate with improved survival of HCC patients [32]. Furthermore, immune scores calculated from gene expression data were used to indicate immune signatures and even estimate the infiltration of immune cells in tumor tissue. In our study, based on TCGA data, the clinicopathological information and immune scores of liver cancer patients were used to explore the relationship between immune scores and prognosis. When adjusted for possible confounders, higher immune scores significantly conferred longer survival times among liver cancer patients. The possible reason for this is that higher immune scores indicated an enhanced immune system and function, which could be mobilized to increase the antitumor immunity of tumor microenvironments so as to control and eliminate the tumor [33]. This is also verified by functional analysis of differentially expressed genes with different immune scores.

Cancer immunotherapy has achieved positive clinical responses in the treatment of various cancers, including liver cancer [34,35]. Immune checkpoint inhibitors have emerged as

potentially effective treatments for patients with HCC in the advanced stage [36]. Clinical experience with checkpoint inhibitors in HCC includes early trials with the anti–cytotoxic T-lymphocyte-associated protein 4 agent tremelimumab and a large phase II trial with the anti–programmed cell death protein 1 agent nivolumab. The latter has shown strong activity—particularly as second-line therapy—both in terms of tumor response and patient survival [8,37]. However, immunotherapy of patients with HCC in the advanced stage remains a great challenge, with very few drugs approved. Therefore, immune scores may not only be used as prognostic biomarkers for liver cancer patients but also have potential clinical values in the choice of immunotherapeutic strategy. At the same time, we constructed a nomogram of liver cancer survival time based on clinicopathological characteristics and immune scores.

*CXCL8* and *CXCL1* belong to the CXC chemokine family, which acts as an important multifunctional cytokine to modulate tumour proliferation, invasion and migration in an autocrine or paracrine manner [38]. *CXCL8* mediated tumor progression, occurring primarily through CXC receptor 1 (*CXCR1*) and CXC receptor 2 (*CXCR2*),has been identified as a function of the modulation of angiogenesis, immune cell infiltration, cell motility, cell survival, and growth in the microenvironment as well as the regulation of local antitumor immune responses [39]. Huang et al. reported that down regulation of *CXCR1* dramatically reduced HCC cell migration, invasion in vitro and lung metastasis in mice model, and HCC patients with positive expression of *CXCL8* or *CXCR1* had shorter overall survival time and higher recurrence rate compared with those with negative expression [40]. *CXCL8* also integrates with multiple intracellular signalling pathways to produce coordinated effects.*PI3K/Akt* pathway is a major downstream signaling pathway of IL-8 inducing cancer cell migration, invasion, and metastasis [41,42].Numerous studies showed that activation of the *PI3K/Akt* signaling pathway was critical to the development and progression of HCC and could modulate the malignant behavior of HCC [43,44]. The *Ras/MAPK* pathway is activated in 50–100% of human HCCs and is related to a poor prognosis [45]. *MAPK* signalling cascade consists of multiple serine/threonine kinases among which the best characterised is the *Raf-1/MAP/Erk* cascade. *CXCL8* activates this classic signalling cascade in both neutrophils and cancer cells [46]. *CXCL1* was regulated by multiple signal pathways and tumor microenvironment. Accumulating evidence has proved that *CXCL1* plays an important role in the development of various malignant tumors. *CXCL1* contributes to tumor-associated neutrophils infiltration in lung cancer which promotes tumor growth [47]. In colon cancer, increased *CXCL1* expression is associated with tumor size, stage, depth of invasion, lymph node metastasis, and patient survival [48]. In this study, we screened out 10 core genes through protein network analysis. By analyzing their relationship with the prognosis of liver cancer, we found the highly expressed *CXCL8* and *CXCL1* are related to the prognosis of liver cancer. Therefore, targeted-inhibition of *CXCL8* or *CXCL1* may be an attractive therapeutic strategy to increase the survival of liver patients. *CXCL8* or *CXCL1* may offer effective approaches for the development of targeted molecular therapeutics for liver cancer.

Some limitations should be noted in our study. Firstly, there was no correlation between survival time and tumor grade because of an insufficient sample size in our analysis. Second, Because of the lack of data about lymph node metastasis in TCGA data in this study, we did not build a clinical prediction model of tumor size, stage and lymph node metastasis for comparison; Third, due to the small number of liver cancer samples in TCGA database, all data samples were in the experimental group and the validation group was not split. Further efforts to collect data relating to immune gene expression and working to incorporate more clinicopathological factors are encouraged to further enhance our models. Also, limited by lack of the treatment information of liver cancer in the TCGA dataset, we were unable to adjust for the

effect of treatment on prognosis. Further research is needed to collect these personal characteristics to improve and verify our models.

Finally, we hope that, with this model, patients and physicians can achieve an individualized survival prediction. Identifying individual subsets that distinguish between different survival risk levels may have an impact on treatment options. Moreover, *CXCL8* or *CXCL1* inhibition warrants further investigation as a candidate therapeutic target in liver cancer.

## Supporting information

**S1 Fig. The database-specific elimination analysis process.**
(TIF)

**S2 Fig. The cutoff points of immune scores by X-tile plotting.** Low immune score subgroup:-1209.2 to -786.4; intermediate immune score subgroup:-786.4 to 268.7; high immune score subgroup:268.7 to 2934.4
(TIF)

**S3 Fig. Kaplan–Meier curves depicting associations of immune score subgroup survival times among patients with liver cancer.** Kaplan–Meier curves depicting that in comparison with patients with low immune scores, those with intermediate and high immune scores had significantly improved survival time. $p < 0.05$; difference was statistically significant.
(TIFF)

**S4 Fig. Multivariate analyses of survival time among liver cancer patients.** Multivariate analyses of survival time among liver cancer patients, when compared with patients with stage I disease, those with stage III or IV disease had significantly poorer survival time; in comparison with patients with low immune scores, those with intermediate and high immune scores had significantly improved survival time. $p < 0.05$; difference was statistically significant.
(TIF)

**S5 Fig. Functional enrichment analysis of differentially expressed immune-related genes.** A. Gene ontology analysis: From top to bottom, the figure represents biological process, cellular component and molecular function, respectively. B. The most significant Kyoto Encyclopedia of Genes and Genomes (KEGG) pathways. The larger the circle, the more genes it contained; conversely, the smaller the circle, the fewer genes it contained. The color of the circle is correlated with the P-value. The smaller the P-value is, the closer it is to the red value. The larger the P-value is, the closer it is to the blue value.
(TIFF)

**S6 Fig. The PPI network and hub genes.** A. PPI network diagram of 33genes. B. The network diagram of top 10 hub genes. PPI–protein-protein interaction
(TIFF)

**S7 Fig. The Overall survival of CXCL8 and CXCL1.** A. Overall survival of CXCL8. B. Overall survival of CXCL1.
(TIFF)

**S8 Fig. Liver cancer survival nomograms.** In these nomograms, each individual patient's value is located on each variable axis and a line is drawn upward to determine the number of points received for each variable value. The sum of these numbers is placed on the total points axis and a line is drawn downward to the survival axes to determine the likelihood of three- or five-year survival.
(TIFF)

**S9 Fig. The calibration curve of survival time at three and five years for liver cancer.**
Nomogram-predicted probability of survival time is plotted on the x-axis; actual survival time is plotted on the y-axis.
(TIFF)

# Acknowledgments

We thank LetPub (www.letpub.com) for its linguistic assistance during the preparation of this manuscript.

# Author Contributions

**Conceptualization:** Qinyan Shen.

**Data curation:** Qinyan Shen.

**Funding acquisition:** Guinv Hu.

**Investigation:** Qinyan Shen.

**Methodology:** Qinyan Shen, Liting Lv.

**Resources:** Qinyan Shen, Liting Lv.

**Software:** Qinyan Shen, Liting Lv.

**Supervision:** Guinv Hu, JinZhong Wu.

**Validation:** JinZhong Wu.

**Visualization:** Qinyan Shen, JinZhong Wu.

**Writing – original draft:** Qinyan Shen.

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
