## [Decision Letter · Decision Letter 0]

23 Apr 2020

PONE-D-20-05913

A new clinical prognostic nomogram for liver cancer based on immune score

PLOS ONE

Dear Mr.Shen Shen,

Thank you for submitting your manuscript to PLOS ONE. After careful consideration, we feel that it has merit but does not fully meet PLOS ONE’s publication criteria as it currently stands. Therefore, we invite you to submit a revised version of the manuscript that addresses the points raised during the review process.

We would appreciate receiving your revised manuscript by Jun 06 2020 11:59PM. To enhance the reproducibility of your results, we recommend that if applicable you deposit your laboratory protocols in protocols.io, where a protocol can be assigned its own identifier (DOI) such that it can be cited independently in the future. For instructions see: http://journals.plos.org/plosone/s/submission-guidelines#loc-laboratory-protocols

We look forward to receiving your revised manuscript.

Kind regards,

Sai-Ching Jim Yeung, MD, PhD

Academic Editor

PLOS ONE

Journal Requirements:

2. Please provide the accession number of URL link to the specific TCGA dataset used in your study.

Additional Editor Comments (if provided):

The nomogram should be applicable to the entire range of possible score. The maximum of total points is 215, not 200.

The font size of the labeling on Figure 3 and Figure 6 should be larger

Reviewers' comments:

Reviewer's Responses to Questions

**Comments to the Author**

1. Is the manuscript technically sound, and do the data support the conclusions?

Reviewer #1: Partly

Reviewer #2: Yes

2. Has the statistical analysis been performed appropriately and rigorously? 

Reviewer #1: No

Reviewer #2: No

3. Have the authors made all data underlying the findings in their manuscript fully available?

Reviewer #1: Yes

Reviewer #2: Yes

4. Is the manuscript presented in an intelligible fashion and written in standard English?

Reviewer #1: Yes

Reviewer #2: Yes

5. Review Comments to the Author

Reviewer #1: This is a study to develop a clinical nomogram using immune score to predict prognosis based on 346 patient sample from TCGA. My following review is with a particular emphasis on the statistical methods and analyses.

1. The proportional hazard assumption for multivariate Cox regression is questionable with crossing survival curves.

2. A comparison to other clinical nomograms without using immune score for liver cancer is needed.

3. It could be risky to apply the proposed nomogram based on the limited sample size in this study without external validation.

Reviewer #2: Overview:

Incidence rates of HCC are growing worldwide with very poor survival. There are few available therapies for HCC, so clearly it is important to develop new ones. QinYan Shen and colleagues have conducted a study that aims to develop a novel prognostic immune nomogram for patients with HCC. They used a published immune score based on the gene expression of stromal and immune cells in tumor samples. They have made the interesting prognostic classifiers (three nomograms) for predict HCC patient survival. They only analyzed the TCGA dataset. Although the prognostic value of immune and/or stromal scores of liver cancer has not been sufficiently investigated, however, the authors did not well describe the methodology, significance and impact of the bioinformatics tool in this HCC study. Limitations of the study include the lack of large validation sample sets to confirm the results obtained from TCGA, and need for more detailed analysis of some specific immune and stromal cell genes, which they pointed out in this study.

Specific comments:

1. Many statements throughout the manuscript lack appropriate references. In introduction section, authors should update most recent HCC clinical and basic research literatures. These should be added where appropriate. The ESTIMATE—a key method they used in this study to evaluate HCC samples but less described in details. Overall, the result section is also too simple and does not focus on what is the significance of the data on HCC study.

2. TCGA also provide normal liver samples either adjacent liver tissues to liver tumor in same patient or nonmalignant liver tissues they collected. It is important to compare the immunoscore not only in HCC but also in nonmalignant liver tissues that may reflect whole liver tumor microenvironment and immunity in liver.

3. It is important to provide and describe what the novelty of the present study is, not only somewhat incremental given that this immunoscore has been shown in other many cancer types.

4. What are the specific immune or stromal makers or immune genes in this HCC data set, such as inflammation cytokines, TGF-beta genes, and exclusive/exhaustive T cell markers? They only show a general correlation between the Score and patient survival in HCC samples. It would strengthen to show some specific immune marker correlations. These data will provide important information for clinical use.

5. Figure legend is too simple. The patients’ information they analyzed and the statistics should be included in. In the Fig.3, what is the detail method they used to get this p value 0.025? is it among 3 groups? Or only low group vs. high/medium groups? It looks like no significant between high and medium?

6. PLOS authors have the option to publish the peer review history of their article (what does this mean?). If published, this will include your full peer review and any attached files.

Reviewer #1: No

Reviewer #2: No

---

## [Author Response · Author response to Decision Letter 0]

16 May 2020

Dear Prof. Sai-Ching Jim Yeung

Thanks a lot for having reviewed our manuscript. Now we have revised the manuscript according to the academic editor and reviewers’ comments. Most of the revisions are in the manuscript.

Response to the academic editor

Question1: Please ensure that your manuscript meets PLOS ONE's style requirements, including those for file naming. The PLOS ONE style templates can be found at https://journals.plos.org/plosone/s/file?id=wjVg/PLOSOne_formatting_sample_main_body.pdf and https://journals.plos.org/plosone/s/file?id=ba62/PLOSOne_formatting_sample_title_authors_affiliations.pdf

Answer1: We ensure that our manuscript meets PLOS ONE's style requirements.

Question2: Please provide the accession number of URL link to the specific TCGA dataset used in your study.

Answer2: We provide the accession number of URL link to the specific TCGA dataset used in our study.

Question3: The nomogram should be applicable to the entire range of possible score. The maximum of total points is 215, not 200. The font size of the labeling on Figure 3 and Figure 6 should be larger

Answer3: We have made changes according to the above requirements of the editor.

Response to Reviewer #1:

Question 1: The proportional hazard assumption for multivariate Cox regression is questionable with crossing survival curves.

Answer1: Compared with the low immune score group, the p value of survival curve of medium or high immune scores group was less than 0.05, the difference was statistically significant. The cross of survival curve is mainly between the medium and high immune scores group, but there is no statistical significance between the medium and high immune scores groups. Kaplan–Meier curves depicting that in comparison with patients with low immune scores, those with medium and high immune scores had significantly improved survival time, so we think that the proportional hazard assumption for multivariate Cox regression is reliable.

Question 2: A comparison to other clinical nomograms without using immune score for liver cancer is needed.

Answer2: In clinic, we usually use tumor size, stage, lymph node metastasis to evaluate the prognosis of patients. Because of the lack of data about lymph node metastasis in TCGA data in this study, we did not build a clinical prediction model of tumor size, stage and lymph node metastasis for comparison. Our study aims to reveal the value of immune score in clinical prediction model and improve the basis for immunotherapy of liver cancer.

Question 3: It could be risky to apply the proposed nomogram based on the limited sample size in this study without external validation.

Answer3: Due to the small number of liver cancer samples in TCGA database, all data samples were in the experimental group and the validation group was not split. This is not only the deficiency of this study, but also the direction of the later study. Later, we plan to build a liver cancer data bank for further verification.

Response to Reviewer #2:

Question 1: Many statements throughout the manuscript lack appropriate references. In introduction section, authors should update most recent HCC clinical and basic research literatures. These should be added where appropriate. The ESTIMATE—a key method they used in this study to evaluate HCC samples but less described in details. Overall, the result section is also too simple and does not focus on what is the significance of the data on HCC study.

Answer1: In introduction section, we add some clinical immune related studies of liver cancer and elaborate the ESTIMATE. In the results section, by analyzing the expression of different genes in the groups with high and low immune scores, we can screen out the target genes that may guide the clinical prognosis of liver cancer to further enrich our results.

Question 2: TCGA also provide normal liver samples either adjacent liver tissues to liver tumor in same patient or nonmalignant liver tissues they collected. It is important to compare the immunoscore not only in HCC but also in nonmalignant liver tissues that may reflect whole liver tumor microenvironment and immunity in liver.

Answer2: As the reviewer said, if there is an immune score of adjacent liver tissues to liver tumor in same patient, it will better reflect the whole tumor microenvironment. Although TCGA database has case data of normal liver tissue, there is no case data of adjacent liver tissues to liver tumor in same patient, and TCGA only has corresponding clinical data of cancer cases; secondly, The ESTIMATE database only has corresponding immune score of cancer patients, so the current data can only reveal immune score of microenvironment of liver cancer.

Question 3: It is important to provide and describe what the novelty of the present study is, not only somewhat incremental given that this immunoscore has been shown in other many cancer types.

Answer3: We further analyzed and screened the immune genes related to the prognosis of liver cancer, and provided theoretical basis for the research of liver cancer clinical targeted drugs.

Question4: What are the specific immune or stromal makers or immune genes in this HCC data set, such as inflammation cytokines, TGF-beta genes, and exclusive/exhaustive T cell markers? They only show a general correlation between the Score and patient survival in HCC samples. It would strengthen to show some specific immune marker correlations. These data will provide important information for clinical use.

Answer4: Through the analysis of differential gene expression in the high and low immune score groups, we screened out 10 target genes, and further analyzed that a and B are related to the prognosis of liver cancer, so as to provide basis for the research and development of clinical targeted drugs for liver cancer.

Question5: Figure legend is too simple. The patients’ information they analyzed and the statistics should be included in. In the Fig.3, what is the detail method they used to get this p value 0.025? is it among 3 groups? Or only low group vs. high/medium groups? It looks like no significant between high and medium?

Answer5: Figure 3 has been modified.

---

## [Decision Letter · Decision Letter 1]

12 Jun 2020

PONE-D-20-05913R1

A new clinical prognostic nomogram for liver cancer based on immune score

PLOS ONE

Dear Dr. Shen,

Thank you for submitting your manuscript to PLOS ONE. After careful consideration, we feel that it has merit but does not fully meet PLOS ONE’s publication criteria as it currently stands. Therefore, we invite you to submit a revised version of the manuscript that addresses the points raised during the review process.

We look forward to receiving your revised manuscript.

Kind regards,

Sai-Ching Jim Yeung, MD, PhD

Academic Editor

PLOS ONE

Additional Editor Comments (if provided):

The responses to the Reviewers' comments were inadequate.

Response to Reviewer#1 Question 1: Reviewer#1 is concerned about the proportional hazard assumption for the Cox regression model. Since Reviewer#1 raised the concern based on the appearance of the Kaplan-Meier curves, the authors should run diagnostic tests for Cox regression model: e.g.,

•Schoenfeld residuals to check the proportional hazards assumption

•Martingale residual to assess nonlinearity.

Response to Reviewer#1 Question 2: The inability to build a clinical prediction model for comparison should be discussed as a limitation of the study in the Discussion section.

Response to Reviewer#1 Question 3: Again, this limitation of small number liver cancer samples is another limitation of the study that should be mentioned in the Discussion section.

Response to Reviewer#2 Question 2: Although the ESTIMATE database only has the corresponding immune scores of the cancer samples, the scores for normal tissue samples can be generated. The TCGA gene expression data for the normal liver samples can be downloaded. The ESTIMATE R packages can be downloaded at https://bioinformatics.mdanderson.org/public-software/estimate/. The ESTIMATE scores can be generated for the normal liver samples and compared.

Reviewers' comments:

Reviewer's Responses to Questions

**Comments to the Author**

1. If the authors have adequately addressed your comments raised in a previous round of review and you feel that this manuscript is now acceptable for publication, you may indicate that here to bypass the “Comments to the Author” section, enter your conflict of interest statement in the “Confidential to Editor” section, and submit your "Accept" recommendation.

Reviewer #2: All comments have been addressed

Reviewer #3: (No Response)

2. Is the manuscript technically sound, and do the data support the conclusions?

Reviewer #2: Yes

Reviewer #3: Partly

3. Has the statistical analysis been performed appropriately and rigorously? 

Reviewer #2: Yes

Reviewer #3: No

4. Have the authors made all data underlying the findings in their manuscript fully available?

Reviewer #2: Yes

Reviewer #3: Yes

5. Is the manuscript presented in an intelligible fashion and written in standard English?

Reviewer #2: Yes

Reviewer #3: Yes

6. Review Comments to the Author

Reviewer #2: (No Response)

Reviewer #3: I am sorry but I don't think the authors have addressed the methodological concerns provided by the previous reviewers.

7. PLOS authors have the option to publish the peer review history of their article (what does this mean?). If published, this will include your full peer review and any attached files.

Reviewer #2: Yes: Jian Chen

Reviewer #3: Yes: Cielito Reyes-Gibby

---

## [Author Response · Author response to Decision Letter 1]

18 Jun 2020

Response to Reviewer #1:

Answer1: Response to Reviewer#1 Question 1: Reviewer#1 is concerned about the proportional hazard assumption for the Cox regression model. Since Reviewer#1 raised the concern based on the appearance of the Kaplan-Meier curves, the authors should run diagnostic tests for Cox regression model: e.g.,

•Schoenfeld residuals to check the proportional hazards assumption

•Martingale residual to assess nonlinearity.

Question 1: Thanks to the reviewer's reminder, the reviewer's intention was not fully understood at the beginning, which resulted in the failure to answer the reviewer's questions well. Now, according to the reviewer's suggestion, we use R language to test the proportional hazard assumption for the Cox regression model. The results show that for age, stage, tumor grade and immune score, P＞0.05.(Figure in Response to Reviewers.docx)

Question 2: Response to Reviewer#1 Question 2: The inability to build a clinical prediction model for comparison should be discussed as a limitation of the study in the Discussion section.

Answer2: We put this part of the deficiency in the discussion part of the article.

Question 3: Response to Reviewer#1 Question 3: Again, this limitation of small number liver cancer samples is another limitation of the study that should be mentioned in the Discussion section.

Answer3: In the discussion part of the article, we will add some explanation on this point.

Response to Reviewer #2:

Question 1: Response to Reviewer#2 Question 2: Although the ESTIMATE database only has the corresponding immune scores of the cancer samples, the scores for normal tissue samples can be generated. The TCGA gene expression data for the normal liver samples can be downloaded. The ESTIMATE R packages can be downloaded at https://bioinformatics.mdanderson.org/public-software/estimate/. The ESTIMATE scores can be generated for the normal liver samples and compared.

Answer1: The tumor microenvironment can be better reflected by the analysis of the immune score of the adjacent tissues of liver cancer. As the reviewer said, the immune score of the normal liver tissue sample data in TCGA can be obtained through the R package, but this part of the normal tissue sample is not the adjacent contrast tissue of the liver cancer tissue.

---

## [Editor Report · Decision Letter 2]

13 Jul 2020

A new clinical prognostic nomogram for liver cancer based on immune score

PONE-D-20-05913R2

Dear Dr. Shen,

We’re pleased to inform you that your manuscript has been judged scientifically suitable for publication and will be formally accepted for publication once it meets all outstanding technical requirements.

Kind regards,

Sai-Ching Jim Yeung, MD, PhD

Academic Editor

PLOS ONE

Additional Editor Comments (optional):

The reviewers' comments have been addressed.
---

## [Editor Report · Acceptance letter]

17 Jul 2020

PONE-D-20-05913R2 

A new clinical prognostic nomogram for liver cancer based on immune score 

Dear Dr. Shen:

I'm pleased to inform you that your manuscript has been deemed suitable for publication in PLOS ONE. Congratulations! Your manuscript is now with our production department. 

Kind regards, 

on behalf of

Dr. Sai-Ching Jim Yeung 

Academic Editor

PLOS ONE